# Mechanically resilient hybrid aerogels containing fibers of dual-scale sizes and knotty networks for tissue regeneration

S. M. Shatil Shahriar[1,2], Alec D. McCarthy[1], Syed Muntazir Andrabi [1], Yajuan Su[1], Navatha Shree Polavoram[1], Johnson V. John[1], Mitchell P. Matis[1], Wuqiang Zhu[3] & Jingwei Xie [1,4] ✉

The structure and design flexibility of aerogels make them promising for soft tissue engineering, though they tend to come with brittleness and low elasticity. While increasing crosslinking density may improve mechanics, it also imparts brittleness. In soft tissue engineering, resilience against mechanical loads from mobile tissues is paramount. We report a hybrid aerogel that consists of self-reinforcing networks of micro- and nanofibers. Nanofiber segments physically entangle microfiber pillars, allowing efficient stress distribution through the intertwined fiber networks. We show that optimized hybrid aerogels have high specific tensile moduli (~1961.3 MPa cm$^3$ g$^{-1}$) and fracture energies (~7448.8 J m$^{-2}$), while exhibiting super-elastic properties with rapid shape recovery (~1.8 s). We demonstrate that these aerogels induce rapid tissue ingrowth, extracellular matrix deposition, and neovascularization after subcutaneous implants in rats. Furthermore, we can apply them for engineering soft tissues via minimally invasive procedures, and hybrid aerogels can extend their versatility to become magnetically responsive or electrically conductive, enabling pressure sensing and actuation.

Aerogels are an innovative category of micro- or nanostructured materials that offer numerous potential biomedical applications, including regenerative medicine, tissue engineering, wound healing, hemostasis, biosensing, wearable electronics, and drug delivery[1–11]. Their distinguishing features, such as ultra-high porosity, lightweight, and flexible design, are particularly appealing in tissue regeneration. However, several significant challenges with current aerogels limit their widespread use in regenerative medicine. These challenges encompass inferior mechanical properties, limited cell infiltration, and slow and partial shape recovery. Consequently, their structural resilience and flexibility are often not ideal, confining their application mainly to largely immobile, non-loadbearing tissues[12].

Attempts to improve mechanical properties have involved increasing initial concentrations and crosslinking densities or optimizing the crosslinking agent, time, and temperature during fabrication. However, these often compromise porosity, pore interconnectivity, and flexibility[6,7,13–15]. This mechanical strength-flexibility conflict has been a persistent issue, hindering the broader application of aerogels. Some strategies have incorporated patterned macrochannels within nanofiber (NF) aerogels using three-dimensional (3D) printed sacrificial templates to promote cellular infiltration[8,16]. However, this method would increase the complexity and cost of fabrication without significantly improving the aerogel's mechanical properties. Also, various materials, such as polyimide[17],

[1]Department of Surgery—Transplant and Mary & Dick Holland Regenerative Medicine Program, College of Medicine, University of Nebraska Medical Center, Omaha, NE 68198, USA. [2]Eppley Institute for Research in Cancer and Allied Diseases, College of Medicine, University of Nebraska Medical Center, Omaha, NE 68198, USA. [3]Department of Cardiovascular Diseases, Physiology and Biomedical Engineering, Center for Regenerative Medicine, Mayo Clinic Arizona, Scottsdale, AZ 85259, USA. [4]Department of Mechanical and Materials Engineering, University of Nebraska Lincoln, Lincoln, NE 68588, USA. ✉e-mail: jingwei.xie@unmc.edu

hydroxyapatite[18], polyurethane[19], gelatin[8,20], cellulose[21,22], and chitin[23] have been explored to fabricate aerogels with 3D fibrillar network structures. While these address the issue of slow and partial shape recovery to some extent, the enhancement in mechanical properties remains modest. Moreover, these aerogels often exhibit non-interconnected pores, small pore sizes, and/or poor mechanical properties (compression modulus: $10^{-2}$–$10^1$ MPa)[4,16,24]. Therefore, developing aerogels that combine high strength, flexibility, and efficient shape recovery while maintaining porosity, pore interconnectivity, and promoting cell infiltration remains a significant challenge.

We hypothesize that the strength-flexibility conflict could be resolved through an aerogel fabrication method that utilizes polymeric NFs and microfibers (MFs) as the constituting materials. Here, soft NFs form a network that intertwines with a more rigid MFs network. During compression, both networks undergo deformation under external stress without breaking because the aerogel allows mechanical stress to be transmitted through the entangled fibrillar network, ensuring shape recovery and preventing mechanical failure[25,26]. The entanglement of fibrillar segments functions as additional binders that cannot be disassembled without breakage in this hybrid aerogel containing fibers with dual-scale sizes[27,28]. This system has the following advantages. The low crosslinking density and fibrillar networks could allow the aerogels to be highly flexible and elastic. The multiscale fibrillar networks can act as mechanical supports, imparting stiffness, and rigidity, while NFs provide a biomimetic morphology, and MFs assist in forming large pores for cell infiltration.

In this work, we introduce a type of hybrid aerogel composed of short NFs and MFs that are intertwined and coated with a binder to achieve both superior mechanical strength and remarkable flexibility. Crosslinking of the binder reinforced the self-entangled, multiscale fibrillar network structures. Our study demonstrates several important findings: (i) the interaction and entanglement between NFs and MFs bolster the mechanical strength and flexibility of hybrid aerogels; (ii) these hybrid aerogels serve as resilient substrate during compressing, storing substantial elastic energy, thus averting mechanical failure under significant mechanical loads; (iii) the inclusion of dual-scale sized fibers is crucial for flexibility and shape recovery, enabling the delivery of hybrid aerogels through instruments like a cannula or catheter; (iv) these aerogels promote tissue ingrowth, extracellular matrix (ECM) formation, and angiogenesis.

## Results

### Processing and structures

We developed hybrid aerogels using a combination of electrospinning, wet spinning, fiber cutting, freeze-casting, and crosslinking as illustrated in Fig. S1. Firstly, we produced two dimensional (2D) NF mats by electrospinning a solution of poly(ε-caprolactone) (PCL) in dichloromethane (DCM)/dimethylformamide (DMF) (Fig. S1a). We chose PCL as raw materials because it has been widely used in many biomedical applications[29]. We then treated the mats with air plasma and cut them into short NFs using cryocutting. Next, we dispersed the short NFs in water to create a short NF suspension via homogenization. Similarly, we fabricated short MF solutions through wet spinning, plasma treatment, cutting, and dispersing (Fig. S1b). The average diameter and length for obtained short NFs were $360.4 \pm 54.2$ nm and $91.5 \pm 14.2$ μm, respectively (Fig. S2a, b). The resulting short MFs had an average diameter and length of $15.5 \pm 1.0$ μm and $1.8 \pm 0.2$ mm, respectively (Fig. S2c, d). We then homogenized the short NFs and short MFs suspended in a 1% gelatin solution in different weight ratios (e.g., 0:100, 25:75, 50:50, 75:25, 100:0), frozen them at −80 °C in customized molds and freeze-dried them to obtain hybrid aerogels. Finally, we crosslinked the hybrid aerogels with 2.5% glutaraldehyde vapor (GA) vapor to stabilize the structure (Fig. S1c).

The hybrid aerogels were produced using a freeze-casting process with customized molds, resulting in various shapes such as cylinder, hollow cylinder, cone, cube (Fig. 1a and Fig. S3)[7]. The ultra-light properties of the aerogels were demonstrated by their ability to be held against gravity by electrostatic force generated between the aerogels (a volume of ~447.7 cm³) and a wooden wall or powder-free hand gloves (Fig. 1b and Supplementary Movie 1). The densities of the hybrid aerogels were significantly lower than those of NF mats and displayed a highly porous structure with porosity of >84%, compared to $8.9 \pm 1.8\%$ for NF mats (Fig. 1c, d). Furthermore, the 2D NF mat exhibited an average pore size of $3.7 \pm 1.4$ μm, whereas the NF aerogel coated with 1% gelatin (NFA) demonstrated a larger size at $60.3 \pm 6.1$ μm (Fig. 1e). Microfiber aerogel coated with 1% gelatin (MFA) displayed the largest average pore size, measuring $265.5 \pm 26.3$ μm, followed closely by NF/MF-A2 (hybrid aerogels containing NF/MF at a ratio of 75:25 w/w coated with 1% gelatin), NF/MF-A1 (hybrid aerogels containing NF/MF at a ratio of 50:50 w/w coated with 1% gelatin), and NF/MF-A3 (hybrid aerogels containing NF/MF at a ratio of 25:75 (w/w) coated with 1% gelatin), with respective sizes of $246.5 \pm 8.1$ μm, $232.1 \pm 11.1$ μm, and $159.3 \pm 10.1$ μm (Fig. 1e). The highly porous structure is ideal for nutrition supply and waste exchange during 3D cell culture, wound healing, and tissue regeneration and fast host cell penetration[30,31]. The scanning electron microscopy (SEM) images revealed the NFAs had a nest-like structures with velvety PCL NFs (Fig. 1f), while MFAs consisted mainly of PCL MFs with a small amount of gelatin nanofibrils (Fig. 1g). Gelatin nanofibrils could be formed due to phase separation, which was similar to the formation of GelMA nanofibrils reported in our recent study[32]. However, differentiating between gelatin nanofibrils and short nanofibers in SEM images is challenging due to their integration and similar appearance. Interestingly, NF/MF-A1 exhibit a bi-continuous fibrillar network where the soft NFs and hard MFs were physically entangled (Fig. 1h). The large pores resulting from the MF component may allow cells to migrate effectively into and within aerogels, while the NF component can mimic the morphology of ECM, making it suitable for 3D cell culture, wound healing, and tissue regeneration applications[33]. We also observed similar morphology and entanglement between NFs and MFs in hybrid aerogels with NF/MF weight ratios of 75:25 (NF/MF-A2) and 25:75 ((NF/MF-A3) supplemented with 1% gelatin (Fig. 1i, j). This entanglement between NFs and MFs may contribute to the superior tear resistance of the hybrid aerogels. Furthermore, the tensor analysis shows that the hybrid aerogels were composed of random fibers, contributing to their isotropic structure, which could be beneficial for tear resistance (Fig. S4a–f).

### Mechanical strength and flexibility

The fabricated hybrid aerogels exhibit high flexibility and excellent compression and bending resistance (Figs. 2a and S5a and Supplementary Movies 2, 3). In addition, hybrid aerogels can rapidly recover their shape in the solutions regardless of pH value (Supplementary Movie 4), while gelatin sponges failed to recover their shape in acidic pH (Fig. S5b–d). This finding indicates the mechanical reinforcement provided by the fibrillar networks. Their shape recovery properties at pH 2.0 demonstrate their potential application in sampling biological specimens in gastrointestinal tract in a minimally invasive manner as described in our recent studies[34]. The shape recovery time of NF/MF-A1 ($1.8 \pm 0.2$ s) is much shorter relative to many other shape-recoverable aerogels and cryogels recently developed, as many described in the literature often require minutes for full recovery (Fig. S5e)[16,23,24,35].

To better understand the mechanical properties, we examined mechanical strength of NFA, MFA, NF/MF-A1, NF/MF-A2, and NF/MF-A3 by performing compression tests at 50%, 70%, and 90% compressive strains (Fig. 2b, c). The compressive strength of aerogels increased with increasing compressive strains. Notably, NFA shows the lowest mechanical resistance among all the tested samples at all the

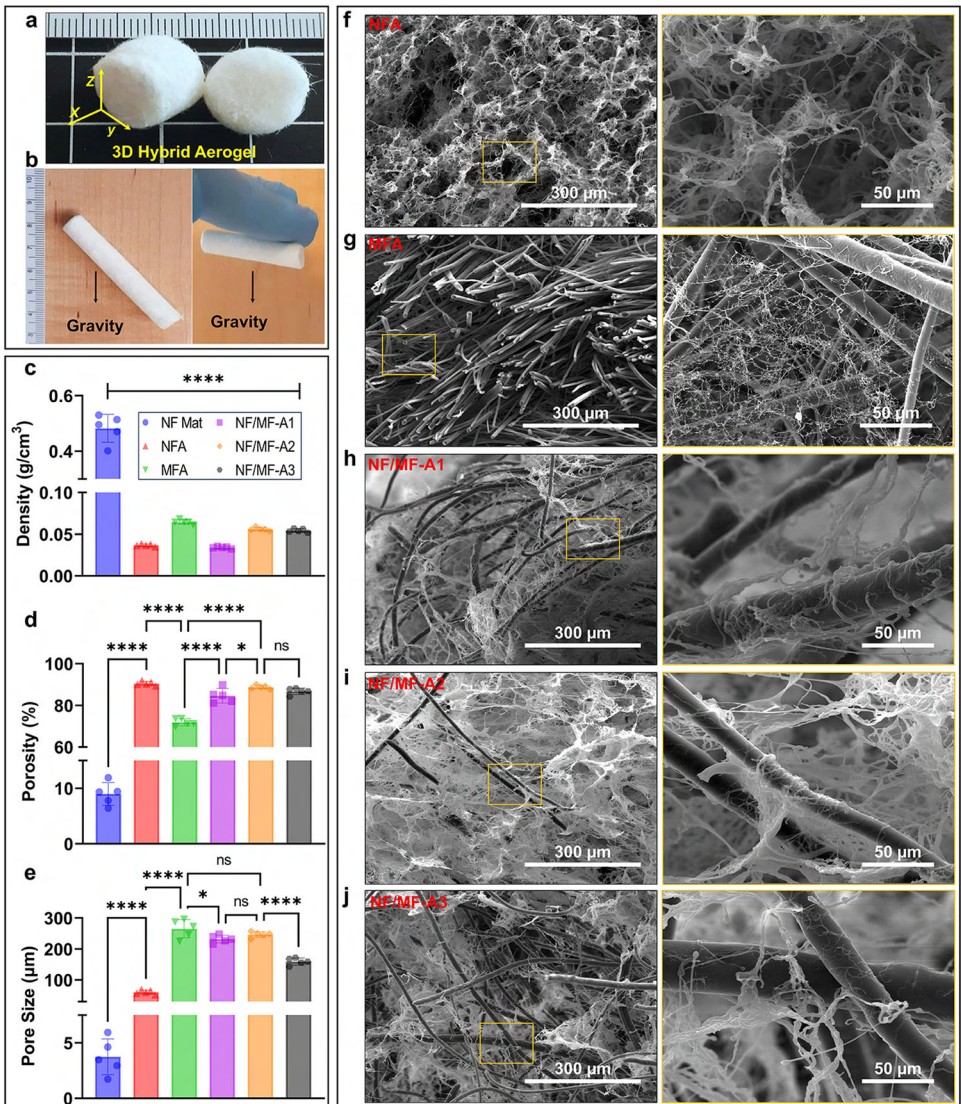

**Fig. 1 | Morphological characterizations of hybrid aerogels. a** Photograph of 3D hybrid aerogels prepared with 50:50 w/w ratio of PCL NF and MF coated with 1% gelatin (NF/MF-A1). **b** Photographs of NF/MF-A1 aerogels supported by a wooden wall or powder-free hand glove against gravity due to electrostatic forces. **c**–**e** Density, porosity, and pore size measurements of samples. Data are presented as mean values ± SD, $N = 5$. The significant difference was detected by one-way ANOVA with Tukey's multiple comparisons test. The 'ns' indicated no significant difference, *$p < 0.01$, ****$p < 0.0001$. **f**–**j** SEM images of NFA, MFA, NF/MF-A1, NF/MF-A2, and NF/MF-A3. Right panel: enlarged images of yellow rectangular areas in the left panel. SEM imaging was repeated three times and similar results were obtained. NFA nanofiber aerogels coated with 1% gelatin. MFA microfiber aerogels coated with 1% gelatin. NF/MF-A1 hybrid aerogels containing NF/MF at a ratio of 50:50 (w/w) coated with 1% gelatin. NF/MF-A2 hybrid aerogels containing NF/MF at a ratio of 75:25 (w/w) coated with 1% gelatin. NF/MF-A3 hybrid aerogels containing NF/MF at a ratio of 25:75 (w/w) coated with 1% gelatin.

compressive strains (Fig. 2c). The mechanical strengths of hybrid aerogels consisting of NF/MF in weight ratios of 25:75, 50:50, and 75:25 dramatically increased compared to NFA and MFA (Fig. 2c, d). NF/MF-A1, NF/MF-A2, and NF/MF-A3 were able to endure the maximum compressive loads of $29.6 \pm 0.6$ MPa, $15.1 \pm 0.1$ MPa, and $23.7 \pm 0.2$ MPa, whereas NFA endures a much smaller strength ($1.7 \pm 0.1$ MPa) under 90% compressive strain (Fig. 2c). Figure 2d shows the excellent resilience of NF/MF-A1, which can endure $8 \pm 0.6$ MPa, $11.1 \pm 1.7$ MPa, and $75.2 \pm 3.6$ MPa of stress under the compressive strains of 50%, 70%, and 90%. NFA shows the least resilience and poorest mechanical robustness, followed by MFA, NF/MF-A2, NF/MF-A3, and NF/MF-A1, suggesting the potential application of hybrid aerogels in structural applications due to their considerable ductility. To further demonstrate the high compression resilience of hybrid aerogels, we performed a cyclic compression-relaxation test (Figs. 2e–g and S6a,b). After ten cycles at 90% strain, NFAs and MFAs

show a ~60% and ~35% decrease in total length due to buckling and plastic deformation under large compressive loads (Fig. 2e, f). Interestingly, during the compression-relaxation cycles at 90% load, compressive stress σ of NF/MF-A1 returned to the original value after unloading for each strain ε. We did not observe any hysteresis loop for NF/MF-A1 during the 100-cycle test (Fig. 2g). Although NF/MF-A3 shows no deformation loss in the total length, NF/MF-A2 shows a ~7% loss under the 90% strain after ten cycles (Figs. S6a, b). These results suggest that the MF network plays an important role in maintaining hybrid aerogels' high elasticity and shape recovery. To test this hypothesis, the high compressive resilience of hybrid aerogels was studied by performing another set of cyclic compression in wet conditions (Figs. S7a–g). Each aerogel underwent five rounds of cyclic compression with each round consisting of three cycles at three different compressive strains (Figs. S7c–g). NFA showed various responses to different strains, with excellent resilience at 50% and 70% and a

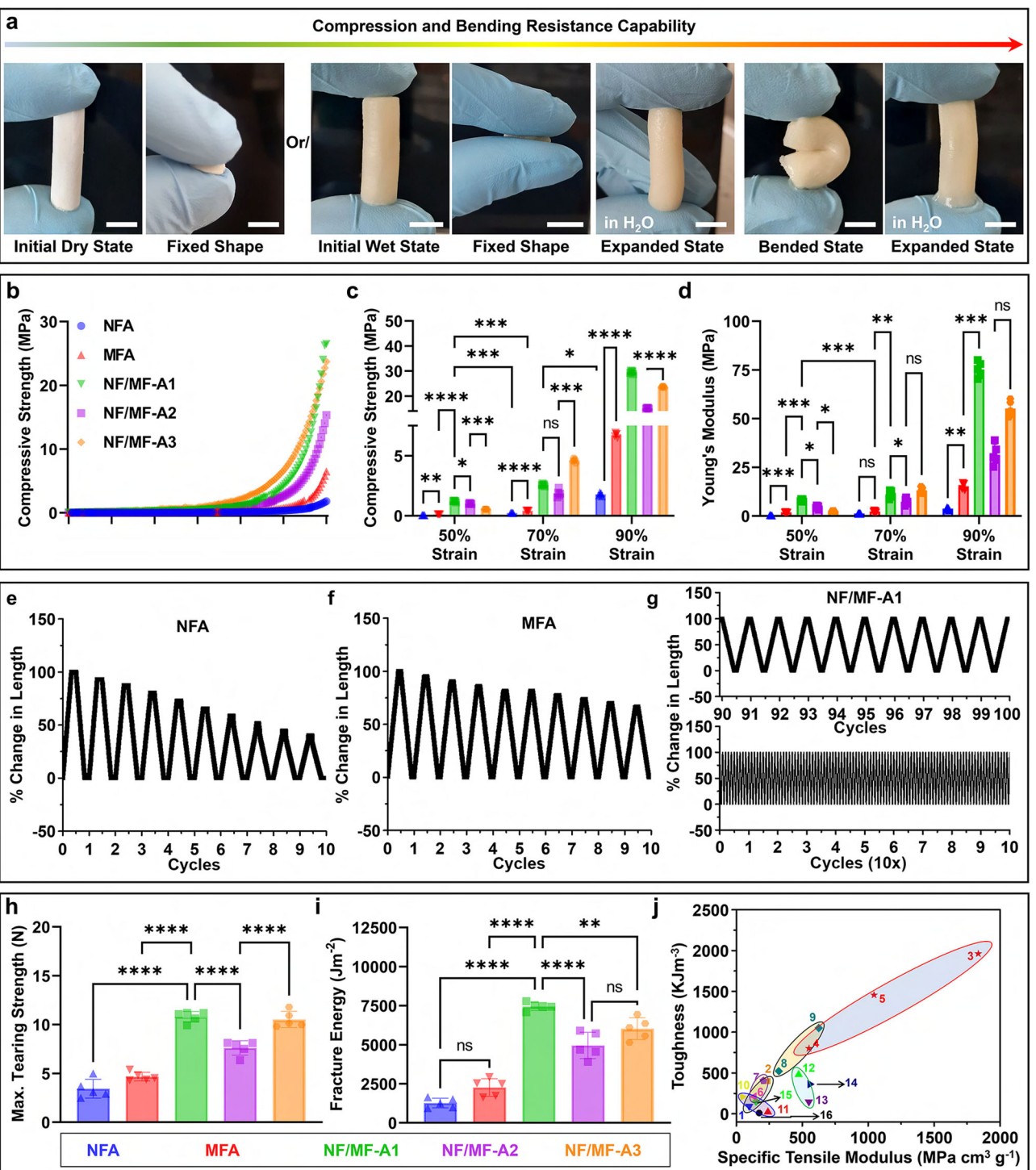

**Fig. 2 | Mechanical characterizations of hybrid aerogels. a** Photographs showing the compression and bending resistance capability of cylindrical hybrid aerogels (NF/MF-A1). Scale = 5 mm. **b** The compression stress-strain curves of NFA, MFA, NF/MF-A1, NF/MF-A2, and NF/MF-A3 at 90% compressive strain. **c**, **d** Maximum compressive strength and Young's modulus at 50, 70, and 90% compressive strains. Data are presented as mean values ± SD, $N = 5$. The significant difference was detected by two-way ANOVA with Tukey's multiple comparisons test. The 'ns' indicated no significant difference, $*p < 0.01$, $**p < 0.001$, $***p \leq 0.0008$, $****p < 0.0001$. **e**–**g**, Changes in length of NFA, MFA, and NF/MF-A1 as a function of their ultra-resilient properties during cyclic compression-relaxation test at 90% compressive strain. **h**, **i** Maximum tearing strength and fracture energies. Data are presented as mean values ± SD, $N = 5$. The significant difference was detected by one-way ANOVA with Tukey's multiple comparisons test. The 'ns' indicated no significant difference, $**p < 0.001$, $****p < 0.0001$. **j** Comparison of mechanical properties of hybrid aerogels with other polymeric aerogels reported in the literature known for high mechanical properties. (1) NFA; (2) MFA; (3) NF/MF-A1; (4) NF/MF-A2; (5) NF/MF-A3; (6) composite nanofiber aerogel with porosity of 91% (CNA-91); (7) composite nanofiber aerogel with porosity of 87% (CNA-87); (8) composite nanofiber aerogel with porosity of 83% (CNA-83); (9) composite nanofiber aerogel with porosity of 76% (CNA-76); (10) bacterial cellulose (BC); (11) BC/ poly(3,4-theylenedioxythiophene)/single walled carbon nanotube (BC/PEDOT/ SWNT); (12) chitin, (13) chitosan, (14) polyimide (PI); (15) PI/carbon nanotube (PI/ CNT); and (16) PI/graphene. NFA nanofiber aerogels coated with 1% gelatin. MFA microfiber aerogels coated with 1% gelatin. NF/MF-A1 hybrid aerogels containing NF/MF at a ratio of 50:50 (w/w) coated with 1% gelatin. NF/MF-A2 hybrid aerogels containing NF/MF at a ratio of 75:25 (w/w) coated with 1% gelatin. NF/MF-A3 hybrid aerogels containing NF/MF at a ratio of 25:75 (w/w) coated with 1% gelatin.

**Table 1 | Toughness and specific tensile modulus of hybrid aerogels compared with other aerogels reported in the literature**

| No. | Aerogels | Porosity (%) | Toughness (KJ m$^{-3}$) | Specific Tensile Modulus (MPa cm$^3$ g$^{-1}$) | Ref. |
|---|---|---|---|---|---|
| 1 | NFA | 96.8 | 92.7 | 98.3 | This work |
| 2 | MFA | 94.3 | 398.2 | 237.1 | |
| 3 | NF/MF-A1 | 95.1 | 1961.3 | 1834.7 | |
| 4 | NF/MF-A2 | 96.4 | 800.5 | 551 | |
| 5 | NF/MF-A3 | 95.3 | 1456.2 | 1045.2 | |
| 6 | CNA-91 | 91 | 212 | 136.4 | 4 |
| 7 | CNA -87 | 87 | 4.6 | 205.6 | |
| 8 | CNA-83 | 83 | 523 | 321.9 | |
| 9 | CNA-76 | 76 | 1050.6 | 625.3 | |
| 10 | BC | - | 207.4 | 45.2 | 45 |
| 11 | BC/ PEDOT/ SWCNT | 76.8 | 22.7 | 240 | 46 |
| 12 | Chitin | 87 | 480.7 | 473.5 | 47 |
| 13 | Chitosan | 83.7 | 147.6 | 550.6 | 48 |
| 14 | PI | 80.3 | 362.3 | 558.3 | 64 |
| 15 | PI/CNT | 59.8 | 152.8 | 149.5 | 44 |
| 16 | PI/ Graphene | 98.9 | 9 | 173.8 | 43 |

*NFA* nanofiber aerogel coated with 1% gelatin, *MFA* microfiber aerogel coated with 1% gelatin, *NF/MF-A1* hybrid NF/MF (50:50 w/w) aerogels, *NF/MF-A2* hybrid NF/MF (75:25 w/w) aerogels coated with 1% gelatin, *NF/MF-A3* hybrid NF/MF (25:75 w/w) aerogels coated with 1% gelatin, *CNA* composite nanofiber aerogel, *BC* bacterial cellulose, *PEDOT* poly(3,4-theylenedioxythiophene), *SWCNT* single walled carbon nanotube, *PI* crosslinked polyimide, *CNT* carbon nanotube.

decrease in compressive resistance at 90% (Fig. S7c). Aerogels consisting of both NF and MF coated with 1% gelatin significantly showed an increase in the mechanical strength (Fig. S7e–g). NF/MF-A1 (NF/MF weight ratio of 50:50 supplemented with 1% gelatin) showed the best compression resilience and highest maximum compression stress (Fig. S7e) among all the aerogels tested, while NF/MF-A2 (NF/MF weight ratio of 75:25 supplemented with 1% gelatin) showed shape recovery loss and mechanical fracture at 90% strain likely due to lower MF content (Fig. S7f). Although NF/MF-A3 (NF/MF weight ratio of 25:75 supplemented with 1% gelatin) showed a lower mechanical resistance among the tested hybrid aerogels, it did not show mechanical fracture at 90% strain (Fig. S7g). These results suggest the crucial role of MFs in maintaining robust mechanical properties for hybrid aerogels. Therefore, resilience and elasticity are largely due to the presence of the NFs, while the major mechanical support is largely imparted by the MFs. The synergism of both structures imparts both favorable mechanical properties to the hybrid aerogels.

To further investigate the improved mechanical properties of hybrid aerogels, we performed tearing tests. The fibrous network in NFA was torn apart under the lowest tearing strength among the tested samples. The tearing strength of hybrid aerogels increased with increasing the MF content (Fig. 2h). For NF/MF-A1, the fibrous networks endured more than three times the tearing strength of NFA because the tension can be readily dissipated by other fibers through entanglements. To illustrate potential applications, we fabricated NF/MF-A1 with human breast and heart shapes (Fig. S8). After a 100-cycle compression-strain test, human breast-shaped NF/MF-A1 showed no deformation or fracture (Supplementary Movie 5). Human heart-shaped NF/MF-A1 showed high flexibility and resilience during 72 cyclic compression-strain tests per minute, similar to a healthy human's heart beating rate (Supplementary Movie 6). We further showed the hybrid aerogels can remain intact under strong mechanical agitation, indicating their structures are stable, which could be mainly attributed to the entanglement between NFs and MFs as well as the crosslinked gelatin component (Fig. S9 and Supplementary Movie 7). Interestingly, NF/MF-A1 had a high fracture strength of 7448.8 ± 228.1 Jm$^{-2}$ which is comparable to the mechanical strength of cardiac or breast tissues (Fig. 2i)[36–42]. The mechanical properties of these hybrid aerogels are superior to previously reported aerogels known as potential candidates for tissue engineering applications in terms of their excellent mechanical properties (Fig. 2j and Table 1)[43–49]. By evaluating the compressive strength, tearing strength, fracture energy, tensile modulus, and toughness of each aerogel, a weighted matrix was employed to choose the most optimized hybrid aerogel in terms of mechanical properties (Supplementary Table S1). Based on the weighted T-Score table (Supplementary Table S2), we chose NF/MF-A1 to for further investigations owing to the most optimal mechanical characteristics.

To understand the mechanism of shape recovery and super elasticity of NF/MF-A1, we performed SEM imaging under 90% compressive strain and after shape recovery (Fig. S10a–c). When compressing wet hybrid aerogels, the absorbed water squeezed out due to the deformation of pores, exhibiting a compact/dense micro- and nano-fibrillar structure (Fig. S10b). The GA crosslinked gelatin and entanglement between fibers can form networks and bridge the fibrillar junctions, thus enabling stress transmission throughout aerogels and preventing the collapse of fibrillar network. Under 90% compressive strain, the hybrid aerogel forms a compressed pellet storing a large amount of elastic potential energy without collapsing or force dissipation. When the compressed pellet in contact with water, it absorbs a large amount of water, and regains its original shape quickly and exerts force due to the release of stored elastic potential energy (Fig. S10c, d). The shaped-recovered aerogels show similar structures to the ones prior to compression (Fig. S10c and Fig. 2g).

## Cell studies

The hybrid aerogels composed of entangled NFs and MFs can recapitulate the natural ECM structure and regulate cell responses. In addition, the highly interconnected porous structure could facilitate cell infiltration and the exchange of nutrition and waste during cell culture. To test this, we first cultured HaCaT cells (immortalized human keratinocytes) on aerogels with different ratios of NFs and MFs (Fig. S11). After 36 h of culture, we performed the live/dead staining of HaCaT cells cultured on either NFA, MFA, or NF/MF-A1. HaCaT cells showed higher proliferation and more uniform distribution on NF/MF-A1 when compared to NFA or MFA, suggesting that NF/MF-A1 may provide a superior environment for cell infiltration and proliferation, which is imparted by both the NFA and MFA (Fig. S11 and Supplementary Movie 8). Due to the fiber size in tens of microns, the pores in MFA were large enough for cell infiltration when seeding cells. However, the seeded cells only interacted with MFs which may be less optimal for cell proliferation relative to NFs. In contrast, because of the fiber size in several hundreds of nanometers, the pore size in NFA was small and thus NFA was not easily penetrated by cells and more seeded cells may be located on the surface layer of NFA. NF/MF-A1 had large pores and NF networks,which could allow cell infiltration during cell seeding, and provide a biomimetic environment for promotion of cell growth. We also cultured green fluorescent protein (GFP)-labled dermal fibroblasts on different aerogels. The GFP-labeled dermal fibroblasts showed a similar trend of sustained and evenly distributed proliferation. It seems that more dermal fibroblasts were seen on NF/MF-A1 at days 3 and 7 after seeding compared to NFA and MFA (Fig. 3a). Importantly, the dermal fibroblasts proliferate throughout the whole hybrid aerogels (Fig. 3b), suggesting their potential application in preparing 3D tissue constructs, rathern than restricting cellular migration to a planar environment.

We also investigated the differentiation of seeded human neural precursor/stem cells on NFA, MFA, and NF/MF-A1 and found neurite

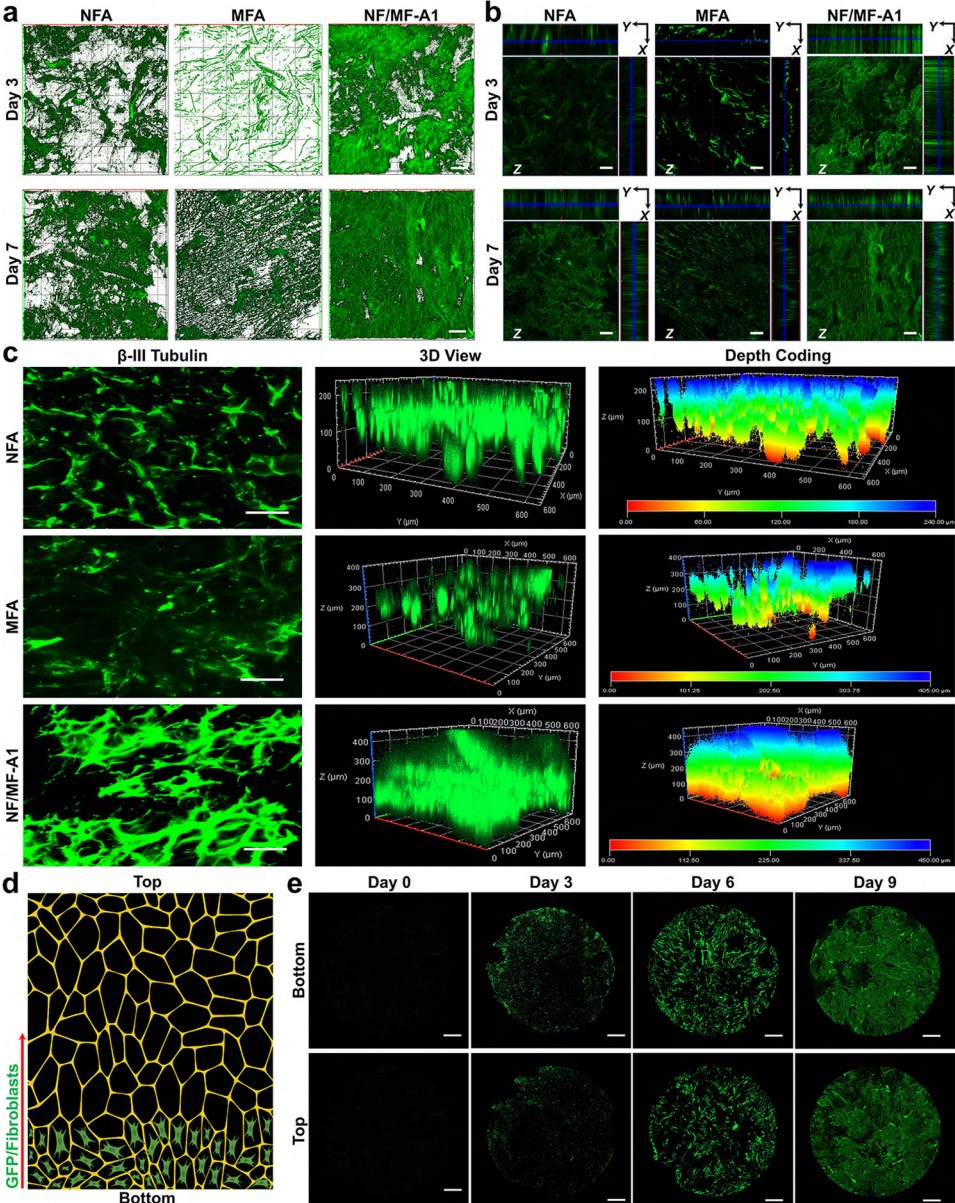

**Fig. 3 | Cell culture. a, b** CLSM images showing 3D and orthogonal view of GFP-labeled dermal fibroblasts in the aerogels after seeding for 3 and 7 days. **c** CLSM images showing human neural progenitor/stem cells cultured on NFA, MFA, and NF/MF-A1 samples for 7 days. Neurons were stained with β-III tubulin in green. 3D view and depth analysis reveal the distribution of neurons/neurites on NFA, MFA, and NF/MF-A1 after 7 days of culture. **d** Schematic illustrating the migration of cells from bottom to top. The schematic was created in BioRender.com. **e** CLSM images showing GFP-labeled dermal fibroblasts on the bottom surface and top surface of NF/MF-A1 aerogels at day 0, 3, 6, and 9. Scale bar = 100 μm for (**a–c**); 1 mm for (**e**). CLSM imaging was repeated three times and similar results were obtained. NFA nanofiber aerogels coated with 1% gelatin, MFA microfiber aerogels coated with 1% gelatin, NF/MF-A1 hybrid aerogels containing NF/MF at a ratio of 50:50 (w/w) coated with 1% gelatin.

outgrowth predominantly observed on the upper surface of NFA, indicating that the initial cell seeding into NFA failed, whereas neurite outgrowth on MFA was significantly deeper, indicating that the cells were able to infiltrate into the deeper location during cell seeding attributed to the larger pores of MFA (Fig. 3c). Hybrid aerogels not only promoted stem cell infiltration but also enhanced neurite outgrowth, which could be due to the large pores and NF networks (Fig. S12). The resultant 3D neural networks on NF/MF-A1 may be useful for modeling neural development and neurological disorders and screening nerve toxic drugs[50].

To delve deeper into the role of NF/MF-A1 in regulating cell proliferation and migration, GFP-fibroblasts were first seeded on petri dishes coated with GelMA. Then, NF/MF-A1 (diameters × height:

8 mm × 2 mm) was positioned atop these cells. The migration of cells from bottom to top was examined by confocal laser scanning microscopy (CLSM) imaging at specific intervals (Fig. 3d). Figure 3e and Fig. S13 show that the area with green fluorescence on the top of the aerogels gradually increased over time, suggesting the migration and proliferation of the GFP-fibroblasts from the bottom to the top surface.

## Minimally invasive delivery

The highly flexible, shape-recoverable properties of hybrid aerogels and the ability for cells to migrate and proliferate rapidly within them suggests their potential applications in delivering functional 3D tissue constructs in a minimally invasive manner[32,51]. To validate this potential, GFP-labeled dermal fibroblasts were seeded on NF/MF-A1 with

different sizes (diameters × height: 8 mm × 2 mm, 10 mm × 2 mm, 12 mm × 2 mm, and 7 mm × 10 mm) (Fig. S14a). After a 14-day culture period, the formed tissue constructs were loaded into an applicator with a 1.6 mm inner diameter cannula and injected into cell culture dishes (Figs. S14b, c). Figure S14c and Supplementary Movie 9 show that such constructs can be delivered through the cannula with a much smaller diameter than the aerogels, which has not been demonstrated for any other type of aerogels. For minimally invasive delivery, it is not necessary to either compress, roll, or fold hybrid aerogel-based tissue constructs inside the applicator unlike the scaffolds previously reported[52–54]. The uncompressed aerogels were able to deform and pass through the cannula likely because of their ultra-flexibility and the applied shear stress during injection. In addition, the composite aerogel created by decellularizing dermal fibroblasts-seeded NF/MF-A1 can be rolled up and regain its shape instantaneously after being delivered through a cannula (Fig. S14d and Supplementary Movie 10). This aerogel showed high elastic modulus and resilience with shape-memory properties after delivery in a minimally invasive manner (Fig. S14e and Supplementary Movie 11). It is worth noting that no significant variation in the cell viability of GFP-labeled dermal fibroblasts was observed on the hybrid aerogels before and after the delivery (Fig. S14f, g).

## Soft tissue regeneration

To explore the potential for tissue regeneration, we implanted 2D NF mats, NFA, MFA, and NF/MF-A1 as cell-free scaffolds into four supraspinal sites on the dorsum of rats and acquired biopsied at 14 and 28 days (Fig. S15a, b). The post-surgical images revealed no severe adverse tissue reactions, such as necrosis, inflammation, or infection, related to the implanted materials at any observed time point (Fig. S15c). A more significant host cell infiltration was seen after 14 days of implantation in NF/MF-A1 and MFA than in 2D NF mats and NFA (Fig. 4a–e). This rapid cell penetration was most likely attributes to the interconnected large pores in NF/MF-A1 and MFA. Such a structure allowed the host cells to migrate from the surrounding tissues to the implants and form new tissues. By 28 days, host cells completely penetrated the NFA, MFA, and NF/MF-A1, forming newly vascularized tissues. Within the same periods, due to the dense structure and small pore size, cells were mainly located on the surface layer of 2D NF mats and failed to penetrate, aligning with findings from earlier studies[55,56]. We further quantified cell infiltration areas and the number of new blood vessels formed in each explant at different time points (Fig. 4f, g). The mean cell infiltrated areas for the 2D NF mats, NFA, MFA, and NF/MFA were 4.1 ± 1.1%, 68.8 ± 5.5%, 92.3 ± 2.8%, 95.7 ± 1.7% after 14 days of implantation (Fig. 4f). The high percentage of cell infiltrated area in NF/MF-A1 within 14 days indicated their great potential applications in tissue regeneration and wound healing. Conversely, by the 28-day mark, the cell infiltrated area reached 96.2 ± 4.3% for NFA. As anticipated, 2D NF mats showed marginal cell infiltration at both intervals. Notably, the NF/MF-A1 displayed a significantly elevated count of new blood vessels compared to both NFA and MFA on days 14 and 28 (Fig. 4g). The blood vessels were evenly distributed throughout the entire newly formed tissues within hybrid aerogels, mirroring the distribution seen in native subcutaneous tissues (Fig. 4c, d).

Furthermore, Masson's trichrome (MT) staining showed enhanced ECM collagen production and neovascularization in the implanted NF/MF-A among all the tested materials on days 14 and 28 (Fig. 5a–c). The collagen deposition area expanded over time across all materials (Fig. 5d), with the highest to lowest amount of collagen deposition being NF/MF-A1, MFA, NFA, and 2D NF mats, respectively (Fig. 5e). The hybrid aerogels had significantly more collagen deposition than NFA ($p < 0.0001$) or MFA ($p < 0.0001$), especially by day 28. This could be due to two distinctly different mechanisms of action. One is that the large-sized microfibers could activate

mechanoreceptors, causing an increase in cell proliferation and ECM synthesis[57]. The other is that the short nanofibers could undergo phagocytosis, which may lead to a subclinical inflammatory response, resulting in macrophage-mediated neocollagenesis[58]. The deposited collagen was homogenously distributed throughout the entire NF/MF-A1, which is akin to the native subcutaneous tissue (Fig. 5a–c).

We further examined the cross-sectional morphologies of each explant on days 14 and 28 using SEM analysis (Fig. S16a–c). The cross-sectional view of 2D NF mat explant revealed densely packed NFs, suggesting that cells failed to penetrate this material due to the compact structure and limited pore size. In stark contrast, the cross-sections of MFA and NF/MF-A1 explants displayed the presence of deposited ECM, red blood cells, and MFs. This observation aligns seamlessly with histological findings that highlighted fast cell infiltration, new blood vessel formation, and ECM deposition throughout the entirety of the NF/MF-A1 in the initial phases (Fig. S16b). Notably, the ECM fibers deposited appear aligned and highly oriented, suggesting the regenerative mechanism results in non-fibrotic neotissue formation (Fig. S16c). NFs were not visible in the cross-sections of NFA and NF/MF-A1 explants, which aligned with the histological observations (Figs. 4, 5, and S16), indicating NFs could be degraded or integrated seamlessly with newly formed tissue and deposited ECM. However, the "void areas" resulting from the presence of microfibers are observable in hematoxylin and eosin (H&E) and MT staining, a finding corroborated by SEM analysis. The hybrid aerogels can also be incorporated with biological cues to further promote host cell recruitment, ECM production, and angiogenesis.

## Functionalization with guest materials

In terms of functionality, hybrid aerogels offer the flexibility to be adapted with specific attributes. For instance, when we coated these hybrid aerogels with polypyrrole (to form NF/MF/Ppy), they exhibited strain-responsive pressure sensing characteristics (Fig. S17a–e). Under compressive strain, the resistance fluctuated but reverted to its original value in each cycle upon release (Figures S17c, d). When compressed, the porous structure's length decreased, shortening the electron transport path and expanding the cross-sectional area available for electrical conduction. This most likely led to reduced electrical resistance (Figure S17e)[50,59]. Owing to their superb shape recovery property and strain-responsive response in a dry state (Figure S17a–e and Supplementary Movie 12), NF/MF/Ppy could be used for pressure sensing and electrical stimulation[60–63]. Additionally, hybrid aerogels enhanced with $Fe_3O_4$ nanoparticles (forming NF/MF/$Fe_3O_4$) responded remarkably to magnetic fields. A magnet could easily control its movement, indicating excellent magnetic responsive properties in both dry and wet conditions (Fig. S17f–h and Supplementary Movie 13). Such magnetic field-responsive hybrid aerogels could be used as actuators or magnetic resonance imaging (MRI)-visible scaffolds for tissue regeneration.

## Discussion

In summary, we have introduced a new class of biomimetic 3D hybrid aerogels crafted through a synergistic approach of electrospinning, wet spinning, freeze-casting, and crosslinking. This combination of chemical crosslinking and physical entanglements within fibrillar networks render these hybrid aerogels remarkable mechanical strength and resilience. These aerogels have a highly porous fibrillar composition, demonstrating resilience against stress and a rapid shape recovery. Notably, NF/MF-A1 can fully regain its shape in less than 2 s, even after withstanding pressures of up to 29.6 ± 0.62 MPa. The robust compression strength of hybrid aerogels caters to the essential equilibrium between strength and flexibility crucial for soft tissue regeneration, particularly in dynamic tissues like the heart and breast, necessitating superior fracture energy and mechanical robustness. Moreover, our hybrid aerogels, characterized by networks of fibers of

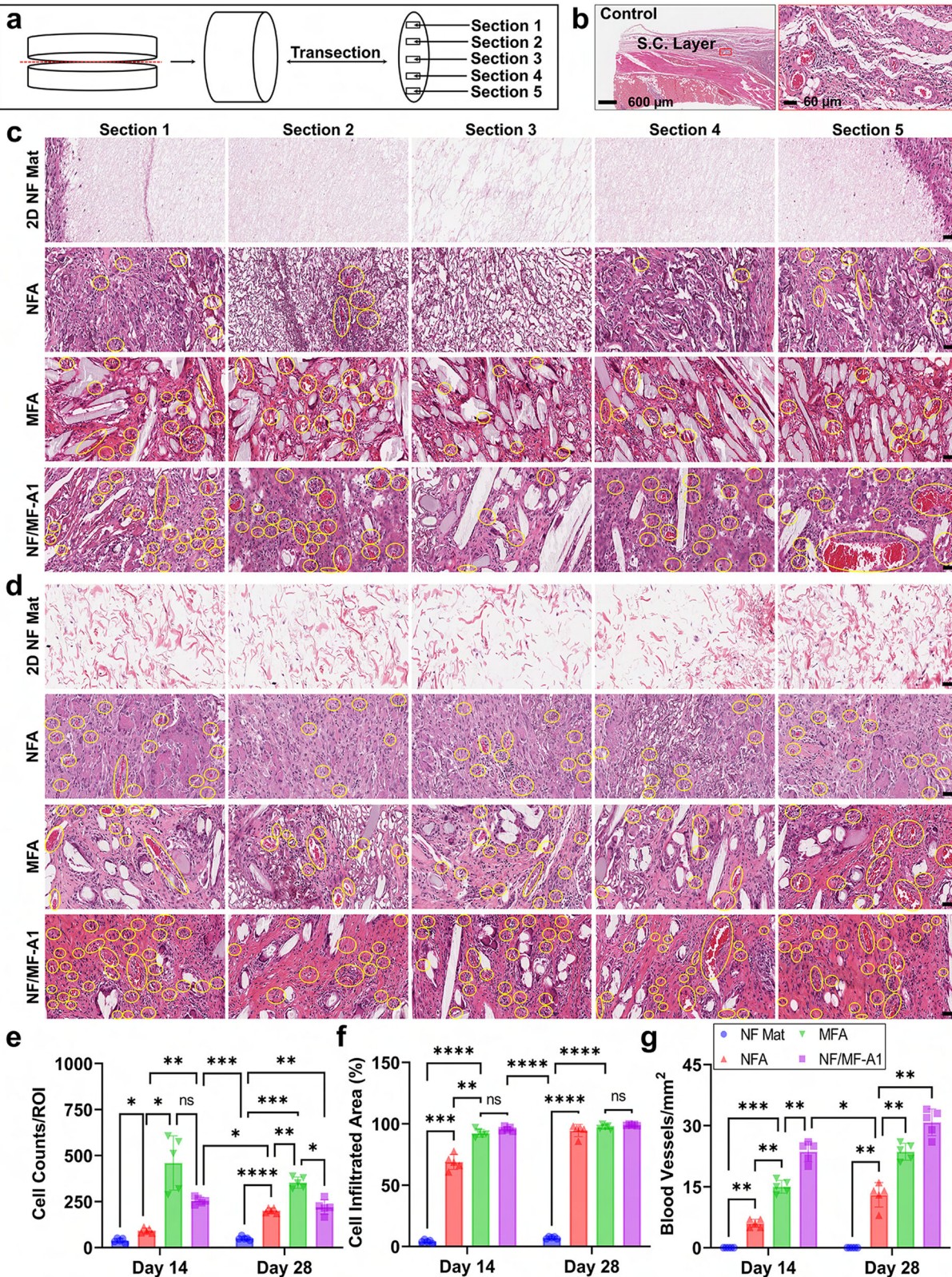

**Fig. 4 | Hybrid aerogels promote cell infiltration and new blood vessel formation after subcutaneous implantation. a** Schematic illustrating different cross-sections of each explant. **b** Hematoxylin and eosin (H & E) staining images of native subcutaneous tissue. **c, d** H & E staining images of different cross-sections of each explant at day 14 and 28. Yellow circles indicate the newly formed blood vessels inside aerogels. Scale bar = 50 μm. Twelve tissue samples were used for histology staining and similar results were obtained. **e** Cell counts per region of interest (ROI) in each explant. **f** Percentages of cell infiltrated area in each explant. **g** Number of newly formed blood vessels in each explant. Data are presented as mean values ± SD, $N = 5$ (randomly selected five tissue samples). The significant difference was detected by two-way ANOVA with Tukey's multiple comparisons test. The 'ns' indicated no significant difference, $*p < 0.01$, $**p < 0.001$, $***p \leq 0.0003$, $****p < 0.0001$. 2D NF mat nanofiber mat, NFA nanofiber aerogels coated with 1% gelatin, MFA microfiber aerogels coated with 1% gelatin, NF/MF-A1 hybrid aerogels containing NF/MF at a ratio of 50:50 (w/w) coated with 1% gelatin.

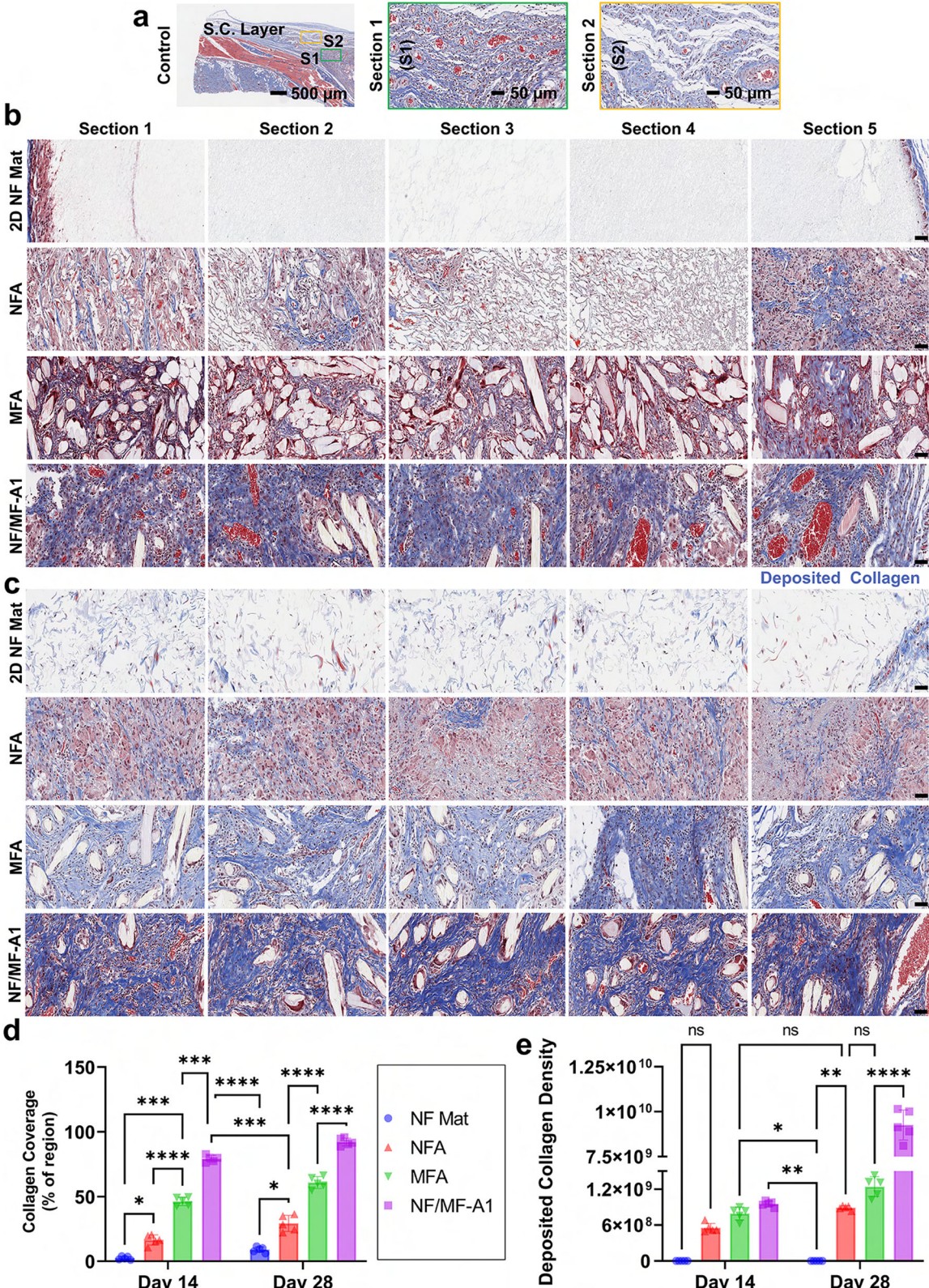

**Fig. 5 | Hybrid aerogels also promote collagen deposition after subcutaneous implantation. a** Trichrome staining images of the native subcutaneous tissue. **b, c** Trichrome staining images of different cross-sections of explants after implantation for 14 and 28 days. The blue color indicates the deposited collagen area. Twelve tissue samples were used for histology staining and similar results were obtained. **d** Percentages of the deposited collagen area in each explant. **e** Quantification of the deposited collagen in each explant. Data are presented as mean values ± SD, $N = 5$ (randomly selected 5 tissue samples). The significant difference was detected by two-way ANOVA with Tukey's multiple comparisons test. The 'ns' indicated no significant difference, $*p < 0.01$, $**p < 0.001$, $***p \leq 0.0006$, $****p < 0.0001$. Scale bar = 50 μm. 2D NF mat nanofiber mat, NFA nanofiber aerogels coated with 1% gelatin, MFA microfiber aerogels coated with 1% gelatin, NF/MF-A1 hybrid aerogels containing NF/MF at a ratio of 50:50 (w/w) coated with 1% gelatin.

dual-scale dimensions, serve as a conducive environment for cell proliferation, migration, and neurite outgrowth in vitro and present an ideal matrix in vivo for fostering host cell infiltration and ECM production. Additionally, the early development of neovascularization facilitates efficient substance exchange, especially in large tissue repair situations. Furthermore, we demonstrated the functionalization of hybrid aerogels with strain-responsive sensing and magnetic-responsive properties. The hybrid aerogels developed in this study could be used for many applications including regeneration of various types of tissues, pressure sensing, actuator, and imaging.

## Methods

### Ethical statement

This research complies with all relevant ethical regulations. All procedures involving animals were done in compliance with National Institutes of Health and Institutional guidelines with approval from the Institutional Animal Care and Use Committee (IACUC) at the University of Nebraska Medical Center (UNMC) (protocol No. 17-103-11-FC).

### Fabrication of hybrid aerogels

The hybrid aerogels were fabricated using a previously unreported technique that combines electrospinning, wet electrospinning, freeze-casting, and crosslinking. The first step was the fabrication of NFs from a NF mat. In brief, 1 g of PCL pellets (Sigma-Aldrich, St. Louis, MO, USA,) were dissolved in a 10 ml of solutions made with 8 ml of DCM (Oakwood Chemical, Estill, SC, USA) and 2 ml of DMF (Oakwood Chemical). An additional 0.005 g of Pluronic F-127 (Sigma-Aldrich) was incorporated into the PCL solution to enhance hydrophilicity. The prepared solution was loaded into a 10 ml syringe for electrospinning. The solution was successively pumped at a flow rate of 0.7 ml/h using a syringe pump (Fisher Scientific, Pittsburgh, PA, USA) and electrospun under a potential of 15-18 kV between the spinneret (21 Gauge needle) using a high voltage generator (ES304-5W/DAM, Gamma High Voltage Research Inc, Ormond Beach, FL, USA). A grounded mandrel (10 cm long and 12 cm in diameter) with the rotating speed of 1200 rpm was used to collect nanofibers. The distance between the needle and collector was 13 cm. The NF mat was removed from the collector by a razor. The NF mat was treated by air plasma and segmented by cryostat cutting. The resultant short NFs were then freeze-dried and kept at 4 °C for further use. The second step was to prepare the short MFs by wet spinning. A 20% w/v PCL solution was prepared by dissolving 4 g PCL pellets in 20 ml of DCM/DMF solution (1/1 v/v). The solution was then loaded into a 20 ml syringe and extruded through a 3D printed 5-emitter extrusion device with 21-gauge needles at a rate of 3.0 ml/h into a coagulation bath of 70% room temperature ethanol using a customized wet spinning device. The multiple emitter device was designed with five outlets and one inlet and was printed using a digital light processing 3D printer (Vida, EnvisionTEC, Gladbeck, Germany) and Clear Guide material (EnvisionTEC). A low-speed mandrel positioned above the ethanol bath was used to collect the solidified MFs. The MF bundle was allowed to dry completely after completing the wet spinning process. The dried MF bundle was cut crosswise using a surgical scissor under liquid nitrogen to avoid pressure-induced fusion of MFs. Finally, short NFs and MFs were treated with air plasma using a High Power Expanded Plasma Cleaner, specifically the PDC-001-HP model, with a voltage rating of 115 V. This equipment was sourced from Harrick Plasma in New York, USA. Subsequently, the fibers were dispersed in water to form suspensions. To prepare short NF and MF suspensions, the dispersed fibers were then homogenized using a probe homogenizer at an amplitude of 20% with 20/10 s on/off cycles for 1 h in ice-cold conditions. Next, short NF and MF suspensions were mixed in the presence of 1% gelatin (derived from bovine skin, Type A, Sigma-Aldrich). This suspension was homogenized for 1 h at the conditions mentioned above. The homogenized suspension was then poured a copper mold glued to an aluminum plate and immediately moved to a −80 °C refrigerator and kept overnight. Upon freeze-drying, the samples were crosslinked under glutaraldehyde vapor (EM grade, 2.5% in Anhydrous Ethanol, Ladd Research, Cincinnati, OH, USA) for 24 h, followed by ethylene oxide gas (Anprolene AN7916 Ethylene Oxide Ampoules, Andersen Sterilizers Inc., Haw River, NC, USA) sterilization.

### Characterization

The dimensions (length and diameters) of short NFs and MFs were calculated using Image J software (NIH). The cross-sectional morphology of hybrid aerogels was observed by SEM (FEI Quanta 200, Hillsboro, OR, USA) at a standard operating condition (accelerating voltage: 25.0 kV, spot: 3.0, and dwell time: 1–3 μs).

The mass and dimensions of cylindrical aerogels were measured using a digital balance and caliper with a resolution of 0.0001 g and 0.01 mm, respectively. The density of aerogels was calculated by the Eq. (1)[1].

$$\rho(\text{g.cm}^{-3}) = \text{m}/\text{V} \tag{1}$$

Where $\rho$ is the density and m and V represent aerogels' mass and volume. The volume of the cylindrical aerogels was calculated by the Eq. (2).

$$V = \pi r^2 h \tag{2}$$

Where r and h represent the radius and height of aerogels, respectively. Additionally, the porosity and pore size of the test samples were assessed using a microCT analysis (Bruker SkyScan 1276 - CMOS Edition, Kartuizersweg, Kontich, Belgium). Next, a structure tensor tool was used to analyze the orientation of NFs and MFs within the hybrid aerogels. The resulting aerogels were frozen in ice and cut using a razor to avoid the fiber deformation during cutting. Each sample was mounted on an SEM imaging stub with silver paste. SEM images were taken at different magnifications for each sample and individual fiber alignment was measured. Distribution of fiber orientation was determined with a structure tensor using a 2-pixel Gaussian window and Gaussian Gradient. The structure tensor at location $x_0$ in the cross-sectional view of hybrid aerogels is defined by the Eq. (3)[2].

$$J(x_0) = \int_{R^2} w(x - x_0)(\nabla f(x))\nabla^T f(x) dx_1 dx_2 \tag{3}$$

where w is a nonnegative isotropic observation window (e.g., a Gaussian) centered at $x_0$. Two measures were defined to estimate the orientation of the fibers by the J: coherence (C) and energy (E). The coherence, C, was computed by the Eq. (4).

$$0 \leq C = \frac{\lambda_{\max} - \lambda_{\min}}{\lambda_{\max} + \lambda_{\min}} = \frac{\sqrt{(J_{22} - J_{11})^2 + 4J_{12}^2}}{(J_{22} - J_{11})} \leq 1 \tag{4}$$

where the eigenvalues of the structure tensor are noted as $\lambda_{\max}$ and $\lambda_{\min}$. Furthermore, the energy of the fibers in the direction θ was computed by the Eq. (5).

$$||D_\theta f||_w^2 = \left(\theta^T \nabla f, \theta^T \nabla f\right)_w = (\theta^T (\nabla f, \nabla f)_w \theta = \theta^T J_\theta \tag{5}$$

Where $||D_\theta f||_w^2$ is the average energy in the window (w).

### Fluid-responsive shape-memory properties

The shape-memory properties of different-shaped aerogels were measured[3]. NFA, MFA, NF/MF-A1, NF/MF-A2, or NF/MF-A3 cylindrical aerogels were manually compressed to achieve the shape-fixed state. Next, the aerogels were put into contact with water or solutions with

different pHs in a shape-fixed state, and their shape-memory properties were recorded by a digital camera (Samsung M30S, Samsung Digital City, Suwon, South Korea). The aerogels' initial dimensions for testing shape-memory in water were 10 mm in diameter and 30 mm in height, whereas, for shape-memory testing in different pH solutions, the dimensions were 10 mm in diameter and 10 mm in height. At different time points, their shape was measured by a digital caliper. In addition to the compression resistance test, their maximum bending resistance was also measured by bending the top part of the aerogels. A digital camera recorded the resistance against external pressure. The external compression release properties of soaked aerogels were also measured by compressing the aerogels (diameter = 10 mm, height = 7 mm) on the top. The amount of released water upon compression and the reabsorbed water upon releasing the force were weighed by a digital balance and recorded by a digital camera (Samsung M30S). The heart-shaped aerogels went through 72 cycles of manual compression and relaxation per min, like healthy human heart which beats 72 times per minute. The breast-shaped aerogels were tested for a 100 manual compression-relaxation cycles to demonstrate their flexibility. The shape-memory properties of the human heart and breast-shaped aerogels were recorded by a digital camera (Samsung M30S).

## Mechanical properties

The aerogels were first examined by a compression test. Aerogels in a cylindrical shape (diameter: 3 mm and height: 10 mm) were tested using an Instron 5640 universal test machine[4]. After fixing the dry samples on the lower compression plate of a CellScale Univert (S/N: UV55290, CellScale Biomaterials Testing, Waterloo, Ontario, Canada) with double-sided tape, a 200 N load plate-initiated compression to reach 50%, 70%, and 90% displacement. To calculate the shape recovery of the fixed samples, all aerogels were soaked in water after 50%, 70%, and 90% compression and their lengths before and after recovery were measured using a digital caliper. Young's modulus was calculated using the Eq. (6)[4–6].

$$\text{Young's Modulus} = \{(F/A)/(\Delta H/H_0)\} = FH_0/A\Delta H \tag{6}$$

Where F, A, $H_0$, and $\Delta H$ represent the compressive force, cross-sectional area, initial height, and change in the height of aerogels, respectively.

For the cyclic compression test, all experiments were carried out in a water bath to prevent water from evaporating from the aerogels[7]. Samples underwent the cyclic compression tests for 10 cycles at 50%, 70%, and 90% strain for each cycle. The durations for compression, hold, recovery, and rest were set for 10, 5, 10, and 5 s for each cycle. It is worth noting that the force was not kept constant during the cyclic compression and relaxation cycles because we measured the force at different strains. Instead, the rate of compression or relaxation was set to 1 mm/s, and force measurements were taken at various strains (i.e., 50%, 70%, and 90%) during compression. However, NF/MF-A1 were tested for 100 cycles to demonstrate their mechanical robustness at a rate of 2 mm/s. The durations for compression, hold, recovery, and rest were set for 5, 1, 5, and 1 s for each cycle. The effect of compressive strains on the mechanical strength of aerogels was tested. The experiment involved subjecting a sample to cyclic compressive strains of 50%, 70%, and 90% for a total of five repetitive rounds. Each round consisted of three repetitive cycles under three different compressive strains, with each cycle including 10 s of compression, 5 s of hold, 10 s of recovery, and 5 s of rest. The sample was initially subjected to the 50% compressive strain, followed by 70% and 90% compressive strains to complete the first round. Subsequently, the same sample was subjected to four more rounds of cyclic compression test under the same condition.

Next, we computed critical forces, max forces, and force loss to determine comparable metrics of compression resistance between aerogels. First, critical force is defined as the force required to initiate compression. We computed critical force as the maximum force reached the top of the force-displacement sliding curve. Similarly, max force is defined as the maximum compressive force achieved during a 90% displacement. Finally, force loss is defined as the percentage difference between max forces in successive cyclic compressions. The change in length for different aerogels after cyclic compression tests is defined as the percentage difference between max changes in height in successive cyclic compressions.

To examine the fracture energy of hybrid aerogels in atmosphere, all tested samples were cut into 50 mm long, 20 mm width, and 4 mm thickness. The two arms of a test sample were clamped, and then the upper arm was pulled upward at a deformation rate of 2 mm/s while the tearing force was recorded. The fracture energy was calculated by the Eq. (7)[7].

$$E = 2F/W \tag{7}$$

Where E is the fracture energy, F is the maximum force from the test, and W is the thickness of the samples[7]. The stability of aerogels in wet conditions were also measured under a strong mechanical agitation (1500 rpm) for 1 h.

## Weighted aerogel ranking

To evaluate the performance of the tested aerogels, a Z-score was computed, and a standardized score was used to transform raw data to make it easier to understand and normalize the range of possible values[4]. The Z-score scale has a mean of 0 and a standard deviation of 1, which allows for comparison across different datasets. Each aerogel's performance was quantified and standardized using the Z-score formula, which subtracts the sample mean ($\mu$) from the observed value ($x$) and divides it by the sample standard deviation ($\sigma$) as shown in the Eq. (8)[4].

$$Z = \frac{x - \mu}{\sigma} \tag{8}$$

To further enhance the analysis, categorical and subjective weighting was applied to several criteria, such as compressive strength, tensile modulus, fracture energy, tearing strength, density, and toughness. By weighing each aerogel's parameter based on its relative importance, the overall performance of each aerogel was evaluated more accurately. The use of Z-scores and standardized weighting allowed for a more objective evaluation of each aerogel's performance. The weights assigned to each criterion were determined based on domain expertize and an intuitive understanding of their relative importance. By assigning standardized weights to the standardized Z-scores, the individual aerogel was ranked based on its performance, allowing for easy identification of the most effective design. This approach allowed for direct comparison among different tested aerogels and provided a clear roadmap for optimizing aerogel design for further study. The weighted average was calculated by the Eq. (9)[8].

$$\bar{x} = \frac{\sum_{i=1}^{n} w_i . x_i}{\sum_{i=1}^{n} w_i} \tag{9}$$

Where x = weighted average, $w_i$ = sum of the product of the weight, and $x_i$ = data number.

## Minimally invasive delivery properties

The minimally invasive delivery of cell-seeded hybrid aerogels and decellularized hybrid aerogels was studied. First, various hybrid aerogels (diameter × height: 8 × 2, 10 × 2, 12 × 2, or 7 × 10 mm²) were first sterilized by the ethylene oxide gas for 12 h. Next, GFP-labeled dermal fibroblasts which were generously supplied by Dr. Mark

Carlson at University of Nebraska Medical Center (UNMC) were seeded on aerogels. After 14 days of culture, the cell-containing aerogel was loaded into an applicator with a 1.7 mm inner diameter cannula for injection. The decellularized aerogels were dried and rolled to test their ability to be delivered in a minimally invasive manner. The injectability of GFP-labeled dermal fibroblast-containing and dry decellularized hybrid aerogels were recorded by a digital camera (Samsung M30S). The viability of the cells in hybrid aerogels before and after injection was also measured using the 3-(4,5-dimethylthiazol-2-yl)-2,5-diphenyl-2H-tetrazolium bromide (MTT) assay (Abcam, Waltham, MA, USA).

## Cell culture studies

Samples were sterilized by ethylene oxide gas for 12 h prior to in vitro cell culture studies. The human immortalized keratinocyte cell line, HaCaT, utilized in assessing cellular toxicity of the tested materials sourced from Antibody Research Corporation (Catalog No. 116027, St. Peters, MO, USA). These cells were stored in a liquid nitrogen tank. Initially, the cells were cultured in Dulbecco's modified Eagle's medium (DMEM) (Gibco Inc., Waltham, MA, USA) containing 10% fetal bovine serum (Abcam, Waltham, MA, USA) and 1% antibiotics (10,000 µg/ml streptomycin and 10,000 units/ml penicillin) (ThermoFisher Scientific, Waltham, MA, USA) at 37 °C with 5% $CO_2$. Next, NFA, MFA, or NF/MF-A1 (diameter = 8 mm, height = 2 mm) were immersed in the complete cell culture media, and $6 \times 10^4$ HaCaT cells were seeded on each aerogel. After 36 h of incubation, cells were stained with ethidium homodimer-1 (Catalog No. L3224, ThermoFisher Scientific) for dead cells in red fluorescence and Calcein-AM (Catalog No. L3224, ThermoFisher Scientific) for live cells in green fluorescence. A CLSM (Zeiss 710, Zeiss, Dublin, CA) was used to obtain the images. To construct the video, z-stake images were merged using Zeiss' Zen Blue software (Zeiss).

GFP-labeled dermal fibroblasts were also seeded on different aerogels and imaged using CLSM. Briefly, $1 \times 10^5$ GFP-labeled dermal fibroblasts were seeded on the NFA, MFA, or NF/MF-A1 and cultured for 3 and 7 days. After culture, the cells on the aerogels were imaged. The 3D view and orthogonal images were directly constructed using Zeiss' Zen Black software (Zeiss) and used without any further modification.

To study the neurite growth, the Human Ren VM NPC cell line (Catalog No. SCC008) was procured from Millipore (Burlington, MA, USA) and cultured in neurobasal media supplemented with 1% MEM non-essential amino acids, 1% $N_2$ supplement, 2% B27 supplement, 1% Glutamax, 0.01% β-mercaptoethanol and 1% penicillin-streptomycin (ThermoFisher Scientific) on Cultrex (R&D systems, Minneapolis, MN, USA)-coated tissue culture plates. $1.0 \times 10^5$ cells were seeded on NFA, MFA, and NF/MF-A1 for 7 days. On day 7, the aerogels containing the cells were washed with DPBS (Gibco Inc.) and fixed in 4% formalin (Sigma-Aldrich) for 5 min. The aerogels were washed twice and blocked with 1% BSA (Sigma-Aldrich) containing 0.1% triton-X 100 (Sigma-Aldrich) in DPBS for 30 min. Primary antibody (Rabbit β-III tubulin, 1:1000 dilution, Catalog No. # ab18207, Abcam, Waltham, MA, USA) was added and incubated at 4 °C overnight. On the next day, the samples were washed thrice with DPBS and incubated with a secondary antibody (Goat anti-rabbit tagged with Alexa fluor 488,1:8000 dilution, Catalog No. # ab150077, Abcam) for 1 h. The samples were then washed thrice and stained with Hoechst 33342 (ThermoFisher Scientific) to visualize the nucleus. Z-stack images were converted to 3D view and depth coding images using Zeiss' Zen Blue software (Zeiss). Depth mapping values were automatically generated by the software and were presented unchanged. The migration of neural stem cells and axonal growth in the aerogels was measured based on 3D view and depth coding images.

To study the migration of cells throughout aerogels, GFP-labeled dermal fibroblasts were first cultured on the culture plate and formed a cell monolayer. NF/MF-A1 was placed on top of the cell monolayer. The migration of cells from the bottom surface to the top surface of aerogels was observed and visualized at different time points (0, 3, 6, and 9 days) using a CLSM (Zeiss LSM800, Zeiss, Dublin, CA)[9]. The cellular migration in the scaffolds was calculated using CLSM images to determine the cell coverage area at different time points[10]. Image J software was used to calculate cell coverage area by measuring the bottom part of the scaffolds (Ab) and the area of cell migration on the top of the aerogel (ACM), which was determined based on the fluorescence of the migrated cells. The percentage of cell coverage area was calculated by the Eq. (10).

$$Cell\ Coverage\ Area\ (\%) = ACm/Ab \times 100 \qquad (10)$$

## Animal studies

The animal study was conducted according to the approved guideline of the IACUC at the UNMC under protocol No. 17-103-11-FC. In brief, 10 weeks old male Sprague Dawley (SD) rats were purchased from Charles River Laboratories (Wilmington, MA, USA), and were housed in the animal facility at UNMC. Rats had free access to regular laboratory food and water and were regularly monitored by staffs in UNMC Comparative Medicine. After 1 week of acclimation facilities, all rats (total = 24) were randomly divided into two cohorts (N = 12) according to the study periods (14 days and 28 days). 1–5% isoflurane through a nose cap was used for anesthesia. Rats were placed on a heating pad to maintain body temperature during the procedure. An area of $8 \times 4\ cm^2$ on the back of each rat was shaved, and the exposed skin was treated with a povidone-iodine solution to create an aseptic environment at the surgical site. Each rat received four different types of implants: 2D NF mat, NFA, MFA, and NF/MF-A1. The dimensions of samples were 10 mm in diameter and 2 mm in height. Incisions of 1.5 cm long were made in four paraspinal sites on the dorsal regions of the SD rats, and one subcutaneous pocket was made on each incision using blunt dissection. Then, the implant was gently inserted through the puncture site into the subcutaneous pocket. After placement of the implants, the skin incisions were closed with staples. The rats were then removed from anesthesia to recover on the heating pad before being transferred to their cage. The rats were euthanized using $CO_2$ inhalation after 14 and 28 days of implantation. The dorsal skin was then carefully resected and immediately immersed in PBS solution. After that, the skin sections containing scaffolds were photographed, and the explants were processed for histological analysis.

## Histology

The isolated explants were submerged in formalin, dehydrated in a graded ethanol series of 70-100%, embedded in paraffin, and finally sectioned at a thickness of 5 µm. H&E and MT stainings were performed by the Tissue Science facility at UNMC. The percentage of cell infiltration area within the explants was analyzed by Image J software. New blood vessel formation was measured by the number of blood vessels per $mm^2$ of explants. The formation of new blood vessels was identified by red blood cells and endothelial lining in the explants. The granulation tissue formation was measured by quantifying the collagen deposition in the explants based on MT staining results. In brief, original MT staining images of explants were converted into RGB images, and these images were deconvoluted by image J software using the color deconvolution plugin. The green channel was identified as collagen fibers. The integrated density of green channels was then quantified[11].

## SEM imaging of dry tissue samples

The cross-sectional morphology of each explant type was characterized by SEM. The chemical-dry method was used to prepare the explant sample for SEM. In brief, the samples were rinsed with 1× DPBS and fixed with a tissue fixative solution made with 2%

paraformaldehyde (PolySciences, Warrington, PA, USA) and 2.5% glutaraldehyde (Ladd Research) in 0.1 M Sorenson phosphate (Electron Microscope Sciences, Hatfield, PA, USA) or sodium cacodylate buffer (Electron Microscope Sciences) for 3 h at room temperature. After fixation, the samples were washed with the same buffer (2×) for 5 min each. Then, the samples were incubated with 1% osmium tetroxide (Sigma-Aldrich) for 30 min at room temperature, followed by three washes with the same buffer (5 min each). The samples were dehydrated in a graded ethanol series (35, 50, 70, 95, and 100%) for 5 min each. The samples were treated in a graded hexamethyldisilazane (HMDS) (Electron Microscope Sciences) series (30, 70, and 100%) for 5 min each. Finally, the samples were left in 100% HMDS to dry in air in the chemical hood.

### Functionalization of hybrid aerogels with polypyrrole and Fe$_3$O$_4$

To demonstrate the hybrid aerogels' versatility for functionalization, we initially outlined the polypyrrole coating method[12]. Briefly, the aerogel (diameter × height: 10 mm × 20 mm) was immersed in a 0.04 M pyrrole (Sigma-Aldrich, St. Louis, MO, USA) solution, mixed with an equal volume of a 0.084 M FeCl$_3$ (Sigma-Aldrich) solution at room temperature, and subjected to 30 min of ultrasonic treatment for uniform polymerization. Subsequently, a 3 V battery-powered circuit was used to measure the initial ($R_0$) and retained ($R$) resistance of the polypyrrole-coated hybrid aerogel (NF/MF/PPy) under compressive strains of 0%, 25%, 50%, 70%, and 90%. The resistance change was quantified using the Eq. (11).

$$\Delta R/R_0 = \frac{R_0 - R}{R_0} \tag{11}$$

Additionally, we introduced Fe$_3$O$_4$ magnetic particles (Millipore Sigma, Burlington, MA, USA), facilitating the integration of magnet-responsive building blocks. Figure S17f displays the schematic illustrating the absorption of Fe$_3$O$_4$ magnetic particles into the hybrid aerogels. Following water removal, the hybrid aerogels maintained their compressed shape, with the subsequent introduction of the Fe$_3$O$_4$ suspension facilitating recovery from deformation. This process led to the development of the NF/MF/Fe$_3$O$_4$ composite hybrid aerogels, and after freeze-drying the composite hybrid hydrogel enabled magnetic attraction through the embedded Fe$_3$O$_4$ particles.

### Statistics and reproducibility

Data were shown as mean ± standard deviation. Statistical analyses were performed using analysis of variance (ANOVA) in GraphPad Prism (version 9.5.1). Pairwise comparisons were made using ordinary one-way ANOVAs with Tukey's multiple comparisons post hoc test. To obtain measurements from photographs/SEM images, Image J was used after calibrating pixels to mm/μm. GraphPad Prism (version 9.5.1), Origin Pro (version 8.5), and BioRender were used to plot and making the graphs/figures.

### Reporting summary

Further information on research design is available in the Nature Portfolio Reporting Summary linked to this article.

## Data availability

All data supporting the findings of this study are available within the article and its supplementary files. Any additional requests for information can be directed to and will be fulfilled by the corresponding author. Source data are provided with this paper.

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

## Acknowledgements

This work was partially supported by startup funds from the University of Nebraska Medical Center (UNMC), National Institute of General Medical Science (NIGMS) and National Heart, Lung, and Blood Institute (NHLBI)

of the National Institutes of Health under Award Numbers R01GM138552 (J.X.) and R01HL162747 (J.X. and W.Z.), Congressionally Directed Medical Research Program (CDMRP)/Peer Reviewed Medical Research Program (PRMRP) FY19 W81XWH2010207 (J.X.), Nebraska Research Initiative grant, and NE LB606.

## Author contributions

S.M.S.S., A.D.M., S.M.A., Y.S., N.S.P., J.V.J., and M.P.M. performed the experiments at UNMC. S.M.S.S., A.D.M., S.M.A., and J.V.J. characterized materials. S.M.S.S., A.D.M., S.M.A., and M.P.M. conducted, analyzed, and interpreted mechanical properties. S.M.S.S., Y.S., and N.S.P. performed cell culture and in vivo studies. W.Z. and J.X. supervised the project. S.M.S.S., A.D.M., and J.X. wrote and revised the manuscript with input from all authors.

## Competing interests

The authors declare no competing interests.
