## [Peer review file · Nature Communications]

REVIEWERS' COMMENTS

Reviewer #1 (Remarks to the Author):

The authors have carried out a very detailed revision of the manuscript, and all main concerns have been fully addressed. Overall, the developed hybrid aerogels have innovative components that provide improved mechanical properties for tissue regeneration. The aerogels have been fully characterized applying a variety of complementary techniques including *in vivo* tests, and the results obtained support the claims of the authors.

Reviewer #2 (Remarks to the Author):

The authors has addressed the concerns raised by the reviewers, and I basically agree to publish the current version in Natural Communications journal. However, I still recommend the authors to use a suitable tissue damage model to evaluate the tissue repair function of the material *in vivo*.

Response to Referees

REVIEWERS' COMMENTS

Reviewer #1 (Remarks to the Author):

The authors have carried out a very detailed revision of the manuscript, and all main concerns have been fully addressed. Overall, the developed hybrid aerogels have innovative components that provide improved mechanical properties for tissue regeneration. The aerogels have been fully characterized applying a variety of complementary techniques including in vivo tests, and the results obtained support the claims of the authors.

Thanks to the reviewer for the encouraging comments.

Reviewer #2 (Remarks to the Author):

The authors has addressed the concerns raised by the reviewers, and I basically agree to publish the current version in Natural Communications journal. However, I still recommend the authors to use a suitable tissue damage model to evaluate the tissue repair function of the material in vivo.

Thanks to the reviewer for the positive comments. The editorial team has decided to overrule the request by Reviewer #2 for in vivo testing of the developed aerogels.